# COVID-19 Publications in Family Medicine Journals in 2020: A PubMed-Based Bibliometric Analysis

**DOI:** 10.3390/ijerph18157748

**Published:** 2021-07-21

**Authors:** Kuang-Yu Liao, Yueh-Hsin Wang, Hui-Chun Li, Tzeng-Ji Chen, Shinn-Jang Hwang

**Affiliations:** 1Department of Family Medicine, Taipei Veterans General Hospital, Taipei 112, Taiwan; yu203209@gmail.com (K.-Y.L.); u9801507@cmu.edu.tw (Y.-H.W.); huinchin@gmail.com (H.-C.L.); sjhwang@vghtpe.gov.tw (S.-J.H.); 2School of Medicine, National Yang Ming Chiao Tung University, Taipei 112, Taiwan

**Keywords:** family medicine, bibliometric analyses, coronavirus disease 2019

## Abstract

Family medicine physicians have been on the front lines of the novel coronavirus disease 2019 (COVID-19) pandemic; however, research and publications in family medicine journals are rarely discussed. In this study, a bibliometric analysis was conducted on COVID-19-related articles published in PubMed-indexed English language family medicine journals in 2020, which recorded the publication date and author’s country and collected citations from Google Scholar. Additionally, we used LitCovid (an open database of COVID-19 literature from PubMed) to determine the content categories of each article and total number of global publications. We found that 33 family medicine journals published 5107 articles in 2020, of which 409 (8.0%) were COVID-19-related articles. Among the article categories, 107 were original articles, accounting for only 26.2% of the articles. In terms of content, the main category was prevention, with 177 articles, accounting for 43.3% of the articles. At the beginning of the epidemic, 10 articles were published in family medicine journals in January 2020, accounting for 11% of all COVID-19-related articles worldwide; however, this accounted for <0.5% of all disciplinary studies in the entire year. Therefore, family medicine journals indeed play a sentinel role, and the intensities and timeliness of COVID-19 publications deserve further investigation.

## 1. Introduction

The novel coronavirus disease (COVID-19), initially reported in late December 2019 at a medical facility in Wuhan, Hubei Province, China, rapidly spread worldwide, posing a serious global health threat [1]. The World Health Organization (WHO) was alarmed by the outbreak and declared it as a global health emergency within one month (30 January 2020) and pandemic on 11 March 2020 [2]. Overall, 80 million confirmed cases and 2 million deaths were recorded by the end of 2020 [3]. This epidemic challenged the global emergency management and medical capacity as well as the research competence for unknown diseases [4]. Until the development of effective treatments and vaccines, disease transmission prevention as well as early detection and diagnosis became one of the most important issues in controlling the epidemic [5,6].

When facing the crisis of the emerging pandemic, all medicine specialties are important. Among them, family medicine physicians play a key role in epidemic prevention and control [7]. Whether in case of severe acute respiratory syndrome (SARS) or influenza in the past or in case of the recent COVID-19, family medicine physicians have been the first line of contact for patients and triage, promoting disease prevention policies during an epidemic [8]. In Singapore, private general practitioner clinics have established information networks to achieve the strategic goal of infection containment and transmission control through government guidelines and logistic support from the health department [9]. Family medicine physicians are crucial for transmission control and early detection of the epidemic.

Since the beginning of the outbreak, an unprecedented worldwide effort has been launched to conduct research on COVID-19 in the face of an unknown novel virus. By May 2021, the number of COVID-19-related publications available on PubMed exceeded 130,000 [10,11]. Owing to this large number of studies, various bibliometric analyses have emerged, including studies from different time periods [12,13], different regions [14], or the most-cited articles and journals [15]. However, although family physicians are on the front line, a bibliometric analysis of COVID-19 conducted from a family medicine perspective is rare.

Therefore, the present bibliometric study was conducted by collecting the publication patterns of global family medicine journals indexed in PubMed in 2020; analyzing the COVID-19-related articles in terms of the number of studies published, their categories, time of publication, and country where the research was conducted; and ranking them according to the most-cited and fastest-published articles. This research and analysis approach provides a further comprehensive view of the contribution of family medicine to emerging infectious disease studies and its key role for combating such viruses.

## 2. Materials and Methods

### 2.1. Why Choose PubMed and Its Disadvantages

Data on COVID-19-related articles were primarily collected from PubMed because it is the largest electronic database that provides free access to biomedical and life science literature. The system is developed and maintained by the National Center for Biotechnology Information (NCBI) at the US National Institute Library of Medicine (NLM), located at the National Institutes of Health. PubMed is linked to several other databases, such as NLM, MEDLINE, and PubMed Central^®^ (PMC) [16]. Moreover, PubMed provides users access to article fragments, such as abstracts of articles, reviews of articles, or options for accessing the full text of publications.

Although PubMed provides a list of all indexed journals, it does not appropriately categorize them according to their professional content and nature. Although there is a classification of Primary Health Care in Broad Subject Terms for Indexed Journals, only journals indexed in MEDLINE are classified. The journals included in PMC are yet not classified. Therefore, we collected family medicine journals in a more objective way using several existing classifications of journals from different databases. Moreover, because English is the most widely spoken language in the world, it has the greatest impact on the promotion of important knowledge. To understand the contribution of journals to the global epidemic, this study focused on English language journals and articles.

### 2.2. Selection of Family Medicine Journals

For selecting family medicine journals, we used the available categories of each database to search for relevant journals. For Science Citation Index (SCI) journals, we used the classification of 2019 InCites Journal Citation Reports in Web of Science. We included 19 journals in the “Primary Health Care” category (Table 1). Non-SCI journals were collected from four databases—NLM, World Organization of Family Doctors (WONCA), Free Medical Journals, and Geneva Foundation for Medical Education and Research (GFMER). From the NLM database, the “Primary Health Care—Includes Family Practice” in the Broad Subject Terms for Indexed Journals was primarily selected, which presented 50 journals. We collected 54 journals from the WONCA website and General Practice/Family Medicine Journals from the official website. For the Free Medical Journals, we selected “Family Practices” from the topics provided and obtained 42 journals. In the GFMER database, we selected the category “Family Medicine, Family Physician, Rural Medicine,” which included 44 journals (Figure 1).

For the collection of non-SCI journals, we filtered the journals collected from different databases in the same way. First, all non-SCI journals were collected; then, all English journals were selected, and the duplicate journals were sorted, resulting in a total of 61 journals, and 29 journals were obtained after searching and comparing with the journal lists provided by MEDLINE [17] and PMC [18]. The status of each journal was reviewed manually, excluding nine journals that have been discontinued and five journals that belong to other disciplines, such as neurology and cardiology, or the Social Science Citation Index. Among them, although the *Asia Pacific Family Medicine* journal was indexed in PubMed, its publication was not included in PubMed in 2020, and the journal will be handled by WONCA in 2021, so it was listed as an irregular publication. Finally, a total of 14 non-SCI family medicine journals were included (Figure 1).

### 2.3. LitCovid: A Database of COVID-19-Related Literature from PubMed 

At the beginning of the COVID-19 epidemic, the Computational Biology Branch of NCBI/NLM developed a new open-resource literature hub called LitCovid, which uses the application programming interface provided by PubMed for linking search tools. It searches PubMed for COVID-19-related literature using the following search terms:

“coronavirus” [All Fields] OR “ncov” [All Fields] OR “cov” [All Fields] OR “2019-nCoV” [All Fields] OR “COVID-19” [All Fields] OR “SARS-CoV-2” [All Fields].

Following this, the results of the search are manually filtered a second time for providing timely access to the scientific literature on the virus biology as well as on diagnostic and treatment practices and patient management. Compared with existing resources, LitCovid provides a further sophisticated search function and visual representation according to the time period (in weeks or days) and geographical location on a world map. Moreover, using manual and advanced machine-learning methods, the articles are divided into eight categories according to their relevance: General Information, Mechanism, Transmission, Diagnosis, Treatment, Prevention, Case Report, and Epidemic Forecasting [19]. To date, LitCovid has collected the most comprehensive international research articles related to COVID-19 (https://www.ncbi.nlm.nih.gov/research/coronavirus/, accessed on 21 July 2021).

### 2.4. Selection of COVID-19-Related Articles

For each article in the included journals in 2020, we searched the official website of each journal, collected, recorded each article from the catalog for each issue in each volume, selected and categorized them. The selection criteria were that the article must be published and indexed in PubMed in 2020 and that it must be in English language. Moreover, we excluded corrections and manually classified the articles into Editorial, Original Article, Short Report, Case Report, Letter, and Others. If the nature of the article is a review but it is classified as an original article by the journal, it would be manually selected and counted as a review. The remaining categories were based on the original classification of each journal; for example, short articles were classified according to the original classification of each journal, and if a case report was classified as a letter, it was counted as a letter. Other article types besides the ones classified above, such as continuing medical education (CME) articles or commentary, were classified as others.

For selecting COVID-19-related articles, we used the list of all articles in the LitCovid collection, i.e., 115,314 articles as of 7 April 2021. There were three details of the article in the list—article title, inclusion year, and abbreviation of the included journals. We first screened the list using the year and obtained 86,499 articles published in 2020. Then, we used the list of family medicine journals included in the previous section to find all articles in these journals and categorized them according to whether they were published in SCI journals or non-SCI journals; we obtained 223 SCI and 276 non-SCI articles. We searched all articles by title in LitCovid and recorded the authors, number of authors, time of PubMed inclusion, and LitCovid article category for each article. Moreover, we excluded non-English articles as well as articles published in PubMed in 2021; in total, there were 25 articles in SCI journals and 65 articles in non-SCI journals. Furthermore, we searched the articles in PubMed and recorded the country of the first author’s affiliation and searched on Google Scholar the number of times each article was cited.

This is a descriptive qualitative analysis study. All data are available as public information and not subject to review according to Taiwan regulations and Institutional Review Board. All data were calculated and analyzed in Microsoft Excel software.

## 3. Results

Overall, 33 family medicine journals, including 34 volumes and 199 issues, were included. Of these, 10 journals (30.3%)—7 SCI journals (36.8% of the SCI group) and 3 non-SCI journals (21.4% of the non-SCI group)—had a specific COVID-19 column (Table 2). These journals published a total of 5107 articles in 2020, of which 2810 articles were published in SCI journals and 2297 articles in non-SCI journals. The most published article type in SCI journals was Others (1156), followed by Original Article (1058), whereas that in non-SCI journals was Original Article (1372), followed by Others (303). In terms of COVID-19-related articles, 409 articles were published in family medicine journals in 2020, accounting for 8.4% of all articles; of these, 198 were published in SCI journals and 211 in non-SCI journals. Regarding classification based on article types, the majority was still Others, with 119 articles accounting for 29.1% of the articles, followed by Original Article, with 107 articles accounting for 26.2% of the articles.

In terms of time period, family medicine journals published 10 COVID-19-related articles as early as January 2020, when the epidemic began to emerge. This accounted for 10.98% of all COVID-19-related articles in PubMed in that month (Figure 2). However, in February and March, there were no relevant published articles. The number of COVID-19-related articles gradually increased until April but was <1% compared with the number of articles published in all disciplines. The number of relevant articles peaked in October with 73 articles, accounting for 17.8% of all the COVID-19-related articles published in family medicine journals for the entire year. Overall, family medicine journals published an average of 34.1 COVID-19-related articles per month.

In 2020, there were 409 COVID-19-related articles published in family medicine journals by 1552 authors from 62 countries (Table 3). The country with the highest number of publications was UK with 76 articles, accounting for 18.6% of all COVID-19-related articles, followed by USA with 58 articles, accounting for 14.2% of all articles. The third country was India with 56 articles, accounting for 13.7% of the articles. China, where the disease originated, published 11 articles, accounting for 2.7%. SCI journals with the most publications were from Australia, with 47 publications, accounting for 23.7% of all SCI journals in family medicine publications. Among the non-SCI journals, journals with the most publications were from India with 52 publications, accounting for 24.6% of all non-SCI journals in family medicine journals. In terms of impact, the country with the highest number of citation times in family medicine journals was UK, with 483 citations, followed by USA, with 231 citations.

Among family medicine journals, the SCI journal that published the most COVID-19-related articles was *Br. J. Gen. Pract.*, with 52 published articles and received the most citations, i.e., 168 citations. The non-SCI journal that published the most COVID-19-related articles was *J. Family Med. Prim. Care*, which published 68 articles and received 135 citations; however, *BJGP Open* received the most citations, with a total of 319 citations. Further, *Can Fam Physician* and *J Prim Care Community Health* were SCI and non-SCI journals that published the fastest COVID-19-related articles.

In terms of the classification based on article content on LitCovid, the most frequent classification was Prevention, with 177 articles (43.3%). The second most popular category was Others, with 159 articles (38.9%). COVID-19-related articles published in family medicine journals in 2020 were those without any epidemic forecasting attribute.

Among the most influential COVID-19-related articles in family medicine journals (Table 4), 11 articles received >30 citations, of which the most-cited articles were classified as Prevention (6 articles), followed by Treatment and Diagnosis (two articles each). Among these 11 articles, 4 were published in *BJGP Open* and received a total of 212 citations, accounting for 66.5% of all COVID-19 citations; these were the only articles in the ranking that were not published in SCI journals.

At the beginning of the outbreak in January, 10 COVID-19-related articles were published in family medicine journals. The first article was published by *Can Fam Physician* on 14 January 2020 (Table 5). The journal with the highest number of published articles was *Br. J. Gen. Pract.* (6 articles); the country with the highest number of publications was the UK. In terms of article content, prevention was the most popular attribute in the fastest publication in January 2020. Among these articles, only two are from non-SCI journals.

## 4. Discussion

In the 33 family medicine journals indexed in PubMed in 2020, 5107 articles were published, 409 of which were COVID-19-related articles. Original articles accounted for approximately one-fourth of the articles in the category. When classified based on article content, articles on prevention accounted for approximately 40% of the articles. At the beginning of the epidemic, family medicine journals published 10 articles in January 2020, accounting for >10% of all COVID-19-related publications worldwide. However, the proportion of COVID-19-related articles published in family medicine journals among all disciplines is extremely low for the entire year.

In 2020, family medicine journals published <0.5% of the total COVID-19-related articles in PubMed worldwide. However, in January 2020, according to the WHO announcement, the number of deaths due to COVID-19 was only 213 [3]. Although the epidemic was not yet considered a worldwide pandemic, there were 10 articles published in family medicine journals, accounting for 11% of the COVID-19-related articles published worldwide at that time. Although none of these articles were original articles of a research nature, the classification of LitCovid shows that they were mainly articles related to prevention. This shows the sensitivity of frontline physicians to emerging diseases and importance of the epidemic in various journals. In February and March 2020, family medicine journals had zero publications on COVID-19. This may be because the epidemic started to get out of control during these two months worldwide. According to the WHO, the number of deaths in March was 36,406 [3], the number of confirmed cases in the USA exceeded 200,000 [20], and following China, the UK entered into a lockdown phase [21]. During this time, the publication of COVID-19 studies worldwide was relatively slow.

Based on the analysis of the number of publications, it was not until April that the number of COVID-19-related articles started to sharply increase, both in family medicine journals and worldwide; however, subsequently, the number of articles published in family medicine journals did not account for >1% of all publications. The reason for this may be that as the severity of the epidemic increased, the number of physicians involved in the frontline for managing the epidemic also increased, thereby reducing the number of family physicians involved in research [22,23]. Furthermore, with the global spread of the epidemic, all research resources were being allocated to the most critical areas, such as virology research, drug therapy, critical care, and vaccines, in an enterprise-like and precise manner [24], consequently reducing the allocation of research resources to other areas. Therefore, it is expected that the proportion of COVID-19-related articles published in family medicine journals is not as high as that in the earlier period.

In terms of country analysis, previous bibliometric studies of COVID-19 have reported a positive correlation between the number of confirmed cases and deaths and the number of COVID-19-related articles published by countries, and China ranked first in terms of both the number of articles published and number of citations, followed by the USA [25]. In the bibliometric analysis of Web of Science, PubMed, and Scopus databases, China ranked first in terms of the number of publications [26,27,28]. Furthermore, in terms of article quality, China was the top publisher of COVID-19-related articles in the top five journals—*Lancet*, *N. Engl. J. Med.*, *Science*, *Nature*, and *JAMA*—in 2020, followed by the USA and UK [29]. Compared with the first SARS outbreak in China in 2003, when the USA and UK were the top publishers [30], China had shown the most prominent research contribution to the outbreak.

However, in the present study, the number of COVID-19-related articles and citations published in family medicine journals in China was considerably lower than those in the UK and USA, with only 11 articles (2.7%) and 71 citations. The main reason is the difference in the emphasis on primary care systems and maturity of development of the country’s healthcare program. In 2009, China started its healthcare reform program, and one of the primary tasks was to develop a community-centered primary healthcare system [31]. However, numerous studies have shown that patients in China choose a more backline hospital for their first visit when they are sick rather than a primary healthcare center [32,33]. Compared with similar studies in the USA and UK, this finding is substantially discrepant [34,35]. Moreover, more than 10 years after the implementation of the reform program, the primary healthcare system in China still has scope for improvement in terms of patient care, noncommunicable disease control, service delivery efficiency, health expenditure control, and public satisfaction [36]. Furthermore, this highlights the fact that research on primary healthcare system requires further attention in China [37].

The 10 fastest-published articles in family medicine journals in 2020 were chiefly published in five journals, among which *BJGP* accounted for six articles. Most journals place great emphasis on COVID-19-related publications; many journals even have specific columns on COVID-19 to ensure that the readers are updated about the latest research information. However, there are journals that do not publish even one COVID-19-related article. There is no single discipline or field of study that is not affected by this worldwide crisis. It is an important responsibility of journals to publish and communicate important knowledge to frontline physicians during the epidemic and the significant impact of manpower reduction, which is one of the most important cornerstones to hinder the progression of the epidemic.

From the past to the present, prevention has been an important part of the training in family medicine [38]. This training has been reflected in combating this viral pandemic. Among the top 10 fastest-published COVID-19-related articles in family medicine journals in 2020, Prevention was the most popular attribute, accounting for 50% of the articles when classified based on content in LitCovid. In terms of the top 11 influential COVID-19-related articles, there were even more articles on prevention, accounting for more than half of the articles. Moreover, on analyzing all articles published in the entire year, prevention accounted for 43.3% of all articles. Among these articles on prevention, although there were several articles related to prevention of COVID-19 transmission and infection, a higher proportion of articles were on prevention of the subsequent issues faced from combating the virus, such as exercise- and nutrition-related issues owing to isolation [39,40], domestic violence issues [41], difficulty faced by students returning to school after being released from isolation [42], psychological issues experienced by families and medical personnel in the face of mass death [43,44], and issues of hospice and palliative care in an epidemic [45]. Although these articles on prevention did not receive a large number of citations, they are vital in terms of the practical aspects of development of the epidemic and its recovery.

In this study, the classification into SCI and non-SCI was conducted mainly for data collection. However, although the number of journals included in the non-SCI group was smaller, the number of COVID-19-related articles published in the non-SCI group was comparable to that of the SCI group, and the total number of citations of COVID-19 articles even exceeded that of the SCI group. This also shows that, although the impact factor or number of citations can provide objective and quantitative data for research, it cannot give a comprehensive evaluation of individual research content.

Considering the unique nature of family medicine, its broad scope, and the overlap with other clinical disciplines, it is relatively difficult to select journals although it was mainly based on the categories available in the original database. However, family medicine physicians often perform work or research beyond these areas and may include other areas, such as geriatrics, hospice medicine, psychiatry, and others. Because a list of all author services that published COVID-19-related articles was unavailable, it was difficult to obtain accurate information on research contributions based on family medicine physicians. Therefore, we opted to base our study on relevant journals. However, a comprehensive study of the contribution of family physicians to the epidemic should be completed by subsequent studies.

During journal selection, we selected five source categories to obtain family medicine journals; however, there are still numerous databases that we did not select. For example, the Broad Subject Terms for Indexed Journals in NLM only includes all MEDLINE-indexed journals and not those indexed in the PMC. The PMC does not have a subject category for journals; accordingly, it is possible that journals essentially on family medicine indexed in the PMC but not in MEDLINE were missed. This results in the underestimating the expression of all family medicine journals on COVID-19.

Regarding the choice of language for journals, English is the most widely spoken language worldwide and thus has the greatest scope for promoting important knowledge. However, owing to the significant increase in work time caused by the epidemic, most frontline physicians in non-English-speaking countries choose medical journals published in their native language to quickly absorb new knowledge about the epidemic. Therefore, we believe that the contribution of non-English language journals to epidemic research is one of the most important issues to be studied in the future.

The timing of publication and PubMed posting may vary among journals. Some articles are published online first before the journal is printed, facilitating the reader to read them on the journal’s website and PubMed. However, some articles are published in the journal itself first and are only included in PubMed after a certain period. We assumed that each reader will search for COVID-19-related articles on family medicine on PubMed and therefore excluded articles published in PubMed until 2021. Therefore, the total number of COVID-19-related articles published in family medicine journals may be underestimated.

Although the articles were mostly classified by the original journal, some of them required manual categorization, and there could easily be different opinions regarding the classification of articles. For example, an article that was classified as a review in the title was actually a CME article. This can be easily overlooked or omitted in the classification, which becomes an unavoidable research limitation.

This is a bibliometric study, and the analysis of the content of individual articles was based on objective quantitative methods, such as numbed of times cited and impact factor of the published journal. However, this method of evaluation could not truly reflect the quality of the research content. Therefore, it could not replace the traditional review articles assessed by experts. For this reason, more review articles are still required to gain a more comprehensive understanding of the research situation in family medicine, such as new research developments in COVID-19 or mainstream issues.

## 5. Conclusions

In this bibliometric study, we determined that family medicine journals demonstrate a high sensitivity to emerging infectious diseases. At the time of the epidemic in January 2020, the number of published articles in these journals accounted for >11% of the articles published worldwide at that time. The main reason for this was the emphasis on disease prevention in family medicine, which was also reflected in the research topic; whether it was the top 10 fastest-published articles, articles with the most citations, or all COVID-19-related articles in 2020, the attribute discussed in these articles was mainly prevention. Among these articles on prevention, besides the prevention of COVID-19 transmission and infection, a higher proportion of articles focused on the prevention of issues that arise from combating the virus and recovery, such as emotional comfort and psychological support for mass mortality. Furthermore, we observed that most family medicine journals have emphasized the publication of COVID-19-related articles, with many of them even creating specific columns on COVID-19. However, there were journals that did not publish even one COVID-19-related article. Among the countries that published articles, the UK and USA were the countries with the highest number of publications and citations. However, compared with other bibliometric studies, China’s research in family medicine journals was not as remarkable.

The epidemic continues to evolve from 2019 to the present. Undoubtedly, the research landscape will continue to change with mutations and spread of the virus globally, and family medicine journals have played a sentinel role at the beginning of the war against the virus. However, the mission of family medicine is not limited to this. As the epidemic is gradually becoming influenza-prone, in addition to prevention, screening and treatment of patients with mild illness and vaccination are important research directions for family medicine in the future. The present study was conducted to understand the state of COVID-19-related publications in family medicine journals and to analyze the contribution and role of family medicine in the epidemic. In addition to the need to strengthen the intensity and continuity of COVID-19-related articles published in these journals, as is the nature of family medicine, there should be more diverse research directions for this disease.

## Figures and Tables

**Figure 1 ijerph-18-07748-f001:**
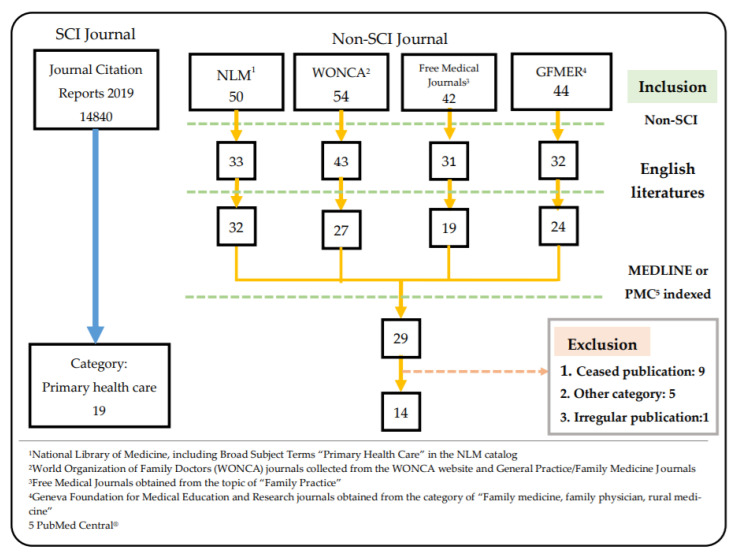
Research method of collection of family medicine journals.

**Figure 2 ijerph-18-07748-f002:**
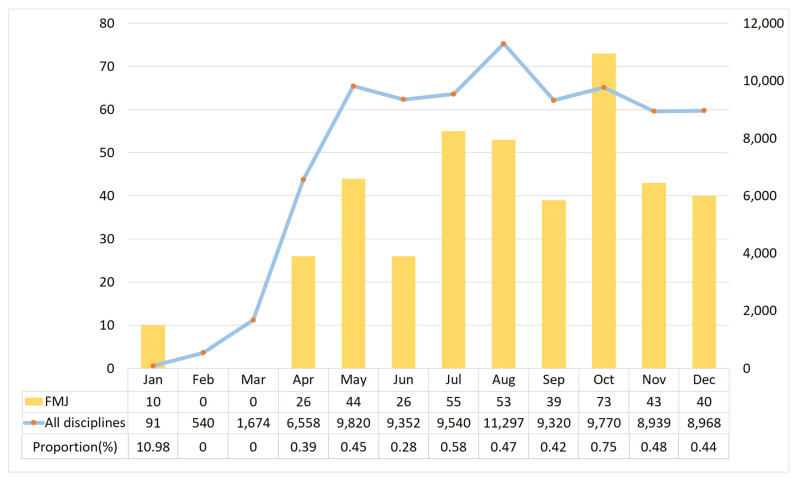
Publication of COVID-19-related articles in family medicine journals in time distribution.

**Table 1 ijerph-18-07748-t001:** All included journals and their PubMed abbreviation.

SCI	SCI Abbr ^1^	Non-SCI	Non-SCI Abbr ^1^
American Family Physician	*Am. Fam. Physician*	African Journal of Primary Health Care and Family Medicine	*Afr. J. Prim. Health Care Fam. Med.*
Annals of Family Medicine	*Ann. Fam. Med*	BJGP Open	*BJGP Open*
Atención Primaria	*Aten. Primaria*	Canadian Journal of Rural Medicine	*Can. J. Rural Med.*
Australian Journal of General Practice	*Aust. J. Gen. Pract.*	Education for Primary Care	*Educ. Prim. Care*
Australian Journal of Primary Health	*Aust. J. Prim. Health*	Family Medicine and Community Health	*Fam. Med. Community Health*
BMC Family Practice	*BMC Fam. Pract.*	Journal of Family and Community Medicine	*J. Family Community Med.*
British Journal of General Practice	*Br. J. Gen. Pract.*	Journal of Family Medicine and Primary Care	*J. Family Med. Prim. Care*
Canadian Family Physician	*Can. Fam. Physician*	Journal of General and Family Medicine	*J. Gen. Fam. Med.*
European Journal of General Practice	*Eur. J. Gen. Pract.*	Journal of Primary Care and Community Health	*J. Prim Care Community Health*
Family Medicine	*Fam. Med.*	Journal of Primary Health Care	*J. Prim. Health Care*
Family Practice	*Fam. Pract.*	Journal of Rural Medicine	*J. Rural Med.*
Journal of Family Practice	*J. Fam. Pract.*	Korean Journal of Family Medicine	*Korean J. Fam. Med.*
Journal of the American Board of Family Medicine	*J. Am. Board Fam. Med.*	Malaysian Family Physician	*Malays Fam. Physician*
npj Primary Care Respiratory Medicine	*NPJ Prim. Care Respir. Med.*	South African Family Practice	*S. Afr. Fam. Pract. (2004)*
Physician and Sports Medicine	*Phys Sportsmed*		
Primary Care: Clinics in Office Practice	*Prim. Care*		
Primary Care Diabetes	*Prim. Care Diabetes*		
Primary Health Care Research and Development	*Prim. Health Care Res. Dev.*		
Scandinavian Journal of Primary Health Care	*Scand. J. Prim. Health Care*		

^1^ Abbreviation in PubMed.

**Table 2 ijerph-18-07748-t002:** General profiles of family medicine journal publication from PubMed in 2020.

		SCI	Non-SCI	Total
Journal (n)		19	14	33
Volume (n)		20	14	34
Issue (n)		132	67	199
Specific COVID-19 column		7	3	10
Article type	n (%)			
	Editorial	158 (5.6)	65 (2.8)	223 (4.3)
	Original Article	1058 (37.6)	1372 (59.7)	2430 (47.5)
	Case Report	26 (0.9)	212 (9.2)	238 (4.7)
	Short	78 (2.8)	46 (2.0)	124 (2.4)
	Review	153 (5.4)	137 (6.0)	290 (5.7)
	Letter	181 (6.4)	162 (7.0)	343 (6.7)
	Others	1156 (41.1)	303 (13.2)	1459 (28.6)
	Total	2810(100)	2297(100)	5107(100)
COVID-19-related article type				
	Editorial	29 (14.6)	14 (6.6)	43 (10.5)
	Original Article	44 (22.2)	63 (29.9)	107 (26.2)
	Case Report	1 (0.5)	3 (1.4)	4 (1.0)
	Short article	1 (0.5)	32 (15.2)	33 (8.1)
	Review	7 (4.5)	30 (14.2)	37 (9.0)
	Letter	33 (16.7)	33 (15.6)	66 (16.2)
	Others	83 (41.9)	36 (17.1)	119 (29.1)
	Total	198 (100)	211 (100)	409 (100)

**Table 3 ijerph-18-07748-t003:** Profiles of COVID-19-related articles in family medicine journals from PubMed in 2020.

COVID-19	SCI	Non-SCI	Total
Journal	19	13	32
Publications	198	211	409
Authors, duplicates allowed	645	907	1552
Countries ^1^	56	46	62
Most productive country (article count)	Australia (47)	India (52)	UK (76)
Numbers of citations, aggregate ^2^	752	988	1740
Most influential country (citation times)	UK (163)	UK (320)	UK (483)
Most productive journal (article count)	*Br. J. Gen. Pract.* (52)	*J. Family Med. Prim. Care* (68)	
Most influential journal (citation times)	*Br. J. Gen. Pract.* (168)	*BJGP Open* (319)	
LitCovid classification, n (%)			
General	4 (2.0)	1 (0.5)	5 (1.2)
Mechanism	0	2 (0.9)	2 (0.5)
Transmission	0	2 (0.9)	2 (0.5)
Diagnosis	15 (7.6)	15 (7.1)	30 (7.3)
Treatment	8 (4.0)	16 (7.6)	24 (5.9)
Prevention	73 (36.9)	104 (49.3)	177 (43.3)
Case Report	6 (3.0)	4 (1.9)	10 (2.4)
Epidemic Forecasting	0	0	0
Others	92 (46.5)	67 (31.8)	159 (38.9)

^1^ The country of the first author’s research affiliation was included. ^2^ Data collected from Google Scholar on 9 April 2021.

**Table 4 ijerph-18-07748-t004:** Top influential COVID-19-related articles in family medicine journals indexed in PubMed in 2020.

Title	Journal (Abbr.)	First Author	PubMed Time	LitCovid Type ^1^	Article Type	Country ^2^	Number of Citations ^3^	SCI
Should Chloroquine and Hydroxychloroquine Be Used to Treat COVID-19? A Rapid Review	*BJGP Open*	Gbinigie, Kome	9 April 2020	Treat	Review	UK	82	
Clinicopathological Characteristics of 8697 Patients with COVID-19 in China: A Meta-Analysis	*Fam. Med. Community Health*	Khan, Moien A B	30 May 2020	Diag	Letter	United Arab Emirates	74	
Telemedicine in the Face of the COVID-19 Pandemic	*Aten. Primaria*	Vidal-Alaball, Josep	14 May 2020	Prev	Original	Spain	67	Yes
The Coronavirus Outbreak: The Central Role of Primary Care in Emergency Preparedness and Response	*BJGP Open*	Dunlop, Catherine	30 January 2020	Prev	Other	UK	66	
Exercise in the Time of COVID-19	*Aust. J. Gen. Pract.*	Fallon, Kieran	24 April 2020	Prev	Other	Australia	39	Yes
COVID-19: Risk of Increase in Smoking Rates among England’s 6 Million Smokers and Relapse among England’s 11 Million Ex-Smokers	*BJGP Open*	Patwardhan, Pooja	9 April 2020	Prev	Other	UK	34	
Family Medicine in Times of “COVID-19”: A Generalists’ Voice	*Eur. J. Gen. Pract.*	de Sutter, An	1 May 2020	Prev	Editorial	Netherlands	32	Yes
Obesity and Risk of COVID-19: Analysis of UK Biobank	*Prim. Care Diabetes*	Yates, Thomas	5 June 2020	Diag	Letter	UK	31	Yes
Physical Distancing with Social Connectedness	*Ann. Fam. Med.*	Bergman, David	13 May 2020	Prev	Original	USA	30	Yes
The COVID-19 Pandemic and Silver Linings for Patient-Centered Care	*Ann. Fam. Med.*	Davis, Ardis	11 November 2020	Other	Other	USA	30	Yes
Should Azithromycin Be Used to Treat COVID-19? A Rapid Review	*BJGP Open*	Gbinigie, Kome	14 May 2020	Treat	Review	UK	30	

^1^ LitCovid article classification to General, Mechanism, Transmission, Diagnosis, Treatment, Prevention, Case Report, and Epidemic Forecasting. Articles not in the abovementioned categories are classified as others. ^2^ The country of the first author’s research affiliation was included. ^3^ Citation times were obtained from Google Scholar on 9 April 2021.

**Table 5 ijerph-18-07748-t005:** Top fastest-published COVID-19-related articles in family medicine journals indexed PubMed in 2020.

Title	Journal (Abbr.)	First Author	PubMed Time	LitCovid Type ^1^	Article Type	Country ^2^	Citation Times ^3^	SCI
Why I Will Not See You on the Barricades: Disability and COVID-19	*Can. Fam. Physician*	Neilson, Shane	14 January 2020	Other	Other	Canada	4	Yes
COVID-19 Highlights Risks of Healthcare and Social Care Workers Attending Work while Ill	*Aust. J. Gen. Pract.*	Hall Dykgraaf, Sally	17 January 2020	Prev	Other	Australia	1	Yes
Five Principles for Pandemic Preparedness: Lessons from the Australian COVID-19 Primary Care Response	*Br. J. Gen. Pract.*	Kidd, Michael R	24 January 2020	Prev	Editorial	Australia	12	Yes
Combating COVID-19: East Meets West	*Br. J. Gen. Pract.*	Li, Donald	24 January 2020	Other	Editorial	China	2	Yes
Dry taps? A Synthesis of Alternative “Wash” Methods in the Absence of Water and Sanitizers in the Prevention of Coronavirus in Low-Resource Settings	*J. Prim. Care Community Health*	Kivuti-Bitok, Lucy W	25 January 2020	Prev	Review	Kenya	2	
COVID-19 Cumulative Mortality Rates for Frontline Healthcare Staff in England	*Br. J. Gen. Pract.*	Levene, Louis S	27 January 2020	Prev	Letter	UK	8	Yes
Domestic Violence during COVID-19: the GP role	*Br. J. Gen. Pract.*	Gibson, Jeremy	27 January 2020	Other	Other	UK	5	Yes
The Atypical Presentation of COVID-19 as Gastrointestinal Disease: Key Points for Primary Care	*Br. J. Gen. Pract.*	Ong, John	27 January 2020	Diag	Review	UK		Yes
Triage of Patients with COVID-19	*Br. J. Gen. Pract.*	Manning, Alex	27 January 2020	Other	Letter	UK		Yes
The Coronavirus Outbreak: The Central Role of Primary Care in Emergency Preparedness and Response	*BJGP Open*	Dunlop, Catherine	30 January 2020	Prev	Other	UK	66	

^1^ LitCovid article classification to General, Mechanism, Transmission, Diagnosis, Treatment, Prevention, Case Report, and Epidemic Forecasting. Articles not in the abovementioned categories are classified as Others. ^2^ The country of the first author’s research affiliation was included. ^3^ Citation times were obtained from Google Scholar on 9 April 2021.

## Data Availability

Data is contained within the article.

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
