# Peer review of "COVID-19 Publications in Family Medicine Journals in 2020: A PubMed-Based Bibliometric Analysis"

_ijerph, 2021, doi:10.3390/ijerph18157748_

Round 1

Reviewer 1 Report

The authors analyse the publications on COVID-19 in family medicine journals worldwide. The first publications appeared shortly after the outbreak of the epidemic but during 2020, the number of publications in family medicine journals was rather low compared to publications in other journals related to COVID-19. In addition, the publications were concentrated in particular countries (UK and US). The importance of the discipline in the fight against newly emerging diseases considered, this might indicate that research and publication capacity of family medicine should be improved.

The study is well conducted and the paper well written.

I think the conclusion should be improved beyond the remark that more research is needed.

At the end of the result section, Line 222-224, results are repeated. You might want to delete these sentences. Or did I understand wrongly? Please check.

Thank you for the opportunity to read this interesting paper.

Author Response

Response to Reviewer 1 Comments

Attachment is the response to all reviewers.

Point 1: The authors analyse the publications on COVID-19 in family medicine journals worldwide. The first publications appeared shortly after the outbreak of the epidemic but during 2020, the number of publications in family medicine journals was rather low compared to publications in other journals related to COVID-19. In addition, the publications were concentrated in particular countries (UK and US). The importance of the discipline in the fight against newly emerging diseases considered, this might indicate that research and publication capacity of family medicine should be improved.

The study is well conducted and the paper well written.

I think the conclusion should be improved beyond the remark that more research is needed.

Response 1: We would like to thank the reviewer for the careful evaluation. We have revised the concluding section. The publication of COVID-19-related articles in family medicine journals should not only be strengthened in terms of intensity and continuity but also should have a more diversified direction for the research topics. Our corrections are as follows:

“The epidemic continues to evolve from 2019 to the present. Undoubtedly, the research landscape will continue to change with mutations and spread of the virus globally, and family medicine journals have played a sentinel role at the beginning of the war against the virus. However, the mission of family medicine is not limited to this. As the epidemic is gradually becoming influenza-prone, in addition to prevention, screening and treatment of patients with mild illness and vaccination are important research directions for family medicine in the future. The present study was conducted to understand the state of COVID-19-related publications in family medicine journals and to analyze the contribution and role of family medicine in the epidemic. In addition to the need to strengthen the intensity and continuity of COVID-19 articles published in these journals, as is the nature of family medicine, there should be more diverse research directions for this disease.”

Point 2: At the end of the result section, Line 222-224, results are repeated. You might want to delete these sentences. Or did I understand wrongly? Please check. Thank you for the opportunity to read this interesting paper.

Response 2: Thank you for the careful review. The results in lines 222–224 are mainly for the top 10 fastest published COVID-19-related articles in January 2020 in family medicine journals. Therefore, they are different from the previous results. However, to avoid confusion among readers, we have revised the paragraph. The revised paragraph is as follows:

 “At the beginning of the outbreak in January, 10 COVID-19-related articles were published in family medicine journals. The first article was published by Can Fam Physician on January 14, 2020. The journal with the highest number of published articles was Br J Gen Pract (six articles); the country with the highest number of publications was the UK. In terms of article content, prevention was the most popular attribute in fastest publication in January 2020. Among these articles, only two are from non-SCI journals.”

______________________________________________________________________________________

Reviewer 2 Report

1.There were some format problems, e.g. Table 2 " Non-SCI " column row 10, the number of digits retained in the percentage should be unified. 2. The meaning of "â…¤" in the "SCI" column should be explained under the table 4. 3. The reasons for excluding " non English " literature in the discussion section can be added to the paragraph describing the selected literature. 4. The relevant background of the Introduction is too detailed, the content should be brief. 5. The proportion of "FMJ " and "All disciplines " cannot be shown by Figure 2.

Author Response

Response to Reviewer 2 Comments

Attachment is the response to all reviewers.

Point 1: There were some format problems, e.g. Table 2 " Non-SCI " column row 10, the number of digits retained in the percentage should be unified.
Response 1: Thank you for the extensive review. The errors in this form have been corrected.
Please see attachment for the correction picture.

Point 2: The meaning of "â…¤" in the "SCI" column should be explained under the table 4.

Response 2: Thank you for the suggestion. It is true that there are many different ways of expressing “Yes” or “Belongs to” in the international context. To avoid misunderstanding, we have changed the term to “Yes” to indicate that the article is published in an SCI journal.
Please see attachment for the correction picture.

Point 3: The reasons for excluding " non English " literature in the discussion section can be added to the paragraph describing the selected literature.

Response 3: Because English is the most widely spoken language worldwide, it has the greatest influence on the promotion of important knowledge. To understand the contribution of each journal to the global epidemic, this study focuses on journals and articles in English for analysis. Based on your comments, we have adjusted the content of the Methods section and provided a sufficient explanation. We have added the following content:

“Although PubMed provides a list of all indexed journals, it does not properly categorize them according to their professional content and nature. Although there is a classification of Primary Health Care in Broad Subject Terms for Indexed Journals, only journals indexed in MEDLINE are classified. The journals included in PMC are yet not classified. Therefore, we collected family medicine journals in a more objective way using several existing classifications of journals from different databases. Moreover, because English is the most widely spoken language in the world, it has the greatest impact on the promotion of important knowledge. To understand the contribution of journals to the global epidemic, this study focused on English language journals and articles.”

Point 4: The relevant background of the Introduction is too detailed, the content should be brief.
Response 4: Thank you for the suggestion. Some descriptive paragraphs have been streamlined to retain the most important parts. The importance of this study has been fully expressed.

Point 5: The proportion of "FMJ " and "All disciplines " cannot be shown by Figure 2.

Response 5: Thank you for the suggestion. Since we relied heavily on Microsoft Excel software for our diagrams, the “proportions” table would not fit smoothly at the bottom if the primary and secondary coordinates were fixed. However, as you suggested, to make it clear for the readers, we have redrawn the diagram and manually added the “proportions” table at the bottom of the drawing.
Please see attachment for the correction picture.

Reviewer 3 Report

  1. The writing of method is very confusing. Suggest clearly state the aims in the introduction as Amis/goals 1,2,3 etc and then describe the method used and results for each aim separately would be very helpful. My comments below show some of the confusions I had when reading the manuscript.
  2. Lines 67-68. Please explain why it is important to conduct such a bibliometric analysis in family medicine. Why knowing “the number of studies published, their categories, the time period of the publication, and the country where the research was conducted; and ranking them according to the most-cited and fastest-published articles” would be important for COVID care in family medicine? I think it would be more important to know what advances in family medicine have been made during COVID regarding caring for COVID patients and what major issues in primary care that are identified in the published COVID-related articles as well the major gaps in the public literature for future studies in family medicine regarding caring for COVID patients.
  3. The justification for exclusion was not provided.
  4. In figure 1, are these number for journals. If so, please explain what “irregular publication” is defined. Also, it is exclude of one journal or just one publication?
  5. At line 88, it stated, “Thereafter, we filtered the journals collected from different databases as follows. First, 19 journals that were included in the 2019 JCR “Primary Health Care” category were 87 excluded”. It’s not clear why these SCI journals were excluded. Then in Table, it seems that these journals were included as Table 1’s tile is “Table 1. All included journals and their PubMed abbreviation”. Please clarify.
  6. It is not clear why SCI is an exclusion criteria. Article in this category could be relevant especially for issues of disparity, access, socioeconomic determines.
  7. The description of article selection is confusing and appears inconsistent. It seems that a PUBMED search was used to identify related articles. For instance, at line 138, it stated “For selecting COVID-19-related articles, we used the list of all articles in the LitCovid 138 collection, i.e., 115314 articles as of April 7, 2021.” There may be journals identified from other databases that are not in PUBMED. Using PUBMED search only may miss articles from these databases. On line 129, it states “The selection criteria were that the article must be published and indexed in PubMed in 2020 and that it must be in the English 129 language.” Then, what is the purpose of identifying relevant journals from other databases in the first place?
  8. There seems to be inconsistency in the description. At line 141, it says, “we used the list of family medicine journals included in the previous section to find all articles in these journals…” There are 33 journals based on Figure 1. Then on page 147, it says, “in total, there were 25 SCI journals and 65 non-SCI journals.” How could that be if only the 33 journals were used to search for articles?
  9. It would be helpful to present the % of publications in each journal that are COVID-related. Only presenting the volume of publications during last year is not very informative.
  10. Line 226, it should be made clear that only publications in PUBMED were assessed.
  11. It would be very helpful to at least identify the major topics in the larger categories of publications such as prevention or diagnosis. Are they merely reporting how many were diagnosed, discussed innovative ways to diagnose patient, or identified issues in diagnosing patients in primary care setting? These would be more informative for improving primary care.

Author Response

Response to Reviewer 3 Comments

Attachment is the response to all reviewers.

Point 1: The writing of method is very confusing. Suggest clearly state the aims in the introduction as Amis/goals 1,2,3 etc and then describe the method used and results for each aim separately would be very helpful. My comments below show some of the confusions I had when reading the manuscript.

Response 1: Thank you for the suggestions, we have reorganized the Methods section to make the process of this study easier for the reader to understand. We have also categorized the entire paragraph according to your suggestion:

2.1 Why Choose PubMed and Its Disadvantages
2.2
Selection of Family Medicine Journals
2.3 LitCovid, A Database of COVID-19-related Literature from PubMed
2.4
Selection of COVID-19-related Articles

Point 2: Lines 67-68. Please explain why it is important to conduct such a bibliometric analysis in family medicine. Why knowing “the number of studies published, their categories, the time period of the publication, and the country where the research was conducted; and ranking them according to the most-cited and fastest-published articles” would be important for COVID care in family medicine? I think it would be more important to know what advances in family medicine have been made during COVID regarding caring for COVID patients and what major issues in primary care that are identified in the published COVID-related articles as well the major gaps in the public literature for future studies in family medicine regarding caring for COVID patients.

Response 2:
Traditionally, bibliometric analysis is a quantitative and descriptive method of presenting the development of research literature on a specific topic in a discipline, region, or country. In the context of the COVID-19 epidemic, as mentioned in the Introduction section, bibliometric studies were conducted in various specialties and countries as a way to understand the investment in COVID-19 research [1-3] . Compared to expert review studies on advance articles, bibliometric studies mostly use the number of publications, impact factor, number of citation and other data to quantify and serve as an objective basis for research investment judgment. Therefore, bibliometric research cannot replace expert review analysis of individual topic publications. Thanks to the reviewer’s comments, we have also added this perspective to our limitation, as follows:

“This is a bibliometric study, and the analysis of the content of individual articles was based on objective quantitative methods, such as number of citations and impact factor of the published journal. However, this method of evaluation could not truly reflect the quality of the research content. Therefore, it could not replace the traditional review articles assessed by experts. For this reason, more review articles are still required to gain a more comprehensive understanding of the research situation in family medicine, such as new research developments in COVID-19 or mainstream issues in research.”

[1] Yang F, Zhang S, Wang Q, et al. Analysis of the global situation of COVID-19 research based on bibliometrics. Health Inf Sci Syst. 2020;8(1):30. Published 2020 Sep 30. doi:10.1007/s13755-020-00120-w
[2] Lee S. Annual report of the productivity and bibliometrics of the Korean Journal of Anesthesiology. Korean J Anesthesiol. 2021;74(1):1-3. doi:10.4097/kja.21015
[3] Maalouf FT, Mdawar B, Meho LI, Akl EA. Mental health research in response to the COVID-19, Ebola, and H1N1 outbreaks: A comparative bibliometric analysis. J Psychiatr Res. 2021;132:198-206. doi:10.1016/j.jpsychires.2020.10.018

Point 3: The justification for exclusion was not provided.

Response 3:
Thank you for the suggestion, which has been explained in the Methods and Materials section for the selection of journals. In the language section, English is the most widely spoken language worldwide and has the greatest influence in the promotion of important knowledge. To understand the contribution of journals to the global epidemic, this study focuses on journals and articles in English. The additional text is as follows:

“Moreover, because English is the most widely spoken language in the world, it has the greatest impact on the promotion of important knowledge. To understand the contribution of journals to the global epidemic, this study focused on English language journals and articles.”

Point 4: In figure 1, are these number for journals. If so, please explain what “irregular publication” is defined. Also, it is exclude of one journal or just one publication?

Response 4:
The numbers in Figure 1 show the number of journals. Among the “irregular publication” journals was Asia Pacific Family Medicine. In 2019, the publisher of the Asia Pacific Family Medicine journal changed from BMC to the Universitas Gadjah Mada Press. However, the articles published in volume 18 (2 issues) in 2020 were not indexed by PubMed. Therefore, it is classified as an irregular publication and excluded. To increase readers’ understanding, we have added a description in Methods:

“The status of each journal was reviewed manually, excluding nine journals that have been discontinued and five journals that belong to other disciplines, such as neurology and cardiology, or SSCI. Among them, although the Asia Pacific Family Medicine journal was indexed in the PubMed, its publication was not indexed in 2020, and the journal will be handled by WONCA in 2021, so it was listed as irregular publication. Finally, a total of 14 non-SCI family medicine journals were included (Figure 1).”

Point 5: At line 88, it stated, “Thereafter, we filtered the journals collected from different databases as follows. First, 19 journals that were included in the 2019 JCR “Primary Health Care” category were 87 excluded”. It’s not clear why these SCI journals were excluded. Then in Table, it seems that these journals were included as Table 1’s tile is “Table 1. All included journals and their PubMed abbreviation”. Please clarify.

Response 5:
Thank you for pointing out that the description of this paragraph can be easily misunderstood. The main purpose of this paragraph was to explain the selection process of the “non-SCI” journals, so 19 journals that had been included in the “SCI” journals were excluded. However, to make the selection process more understandable to the reader, we have redescribed the entire process in terms of the inclusion of non-SCI journals, with the following changes to the text:

“For the collection of non-SCI journals, we filtered the journals collected from different databases in the same way. First, all non-SCI journals were collected; then, all English journals were selected, and duplicate journals were sorted, resulting in a total of 61 journals, and 29 journals were obtained after searching and comparing with the journal lists provided by MEDLINE and PMC. The status of each journal was reviewed manually, excluding nine journals that have been discontinued and five journals that belong to other disciplines, such as neurology and cardiology or SSCI. Among them, although the Asia Pacific Family Medicine journal was indexed in the PubMed, its publication was not included in PubMed in 2020, and the journal will be handled by WONCA in 2021, so it was listed as an irregular publication. Finally, a total of 14 non-SCI family medicine journals were included (Figure 1).”

Point 6: It is not clear why SCI is an exclusion criteria. Article in this category could be relevant especially for issues of disparity, access, socioeconomic determines.

Response 6:
SCI was not an exclusion, as described above and in Methods 2.2 Selection of Family Medicine Journals. In this study, all 19 SCI journals published in the 2019 JCR in the Primary Health Care classification were included in the SCI journal group.

“For selecting family medicine journals, we used the available categories of each database to search for relevant journals. For Science Citation Index (SCI) journals, we used the classification of 2019 InCites Journal Citation Reports (JCR) in Web of Science. We included 19 journals in the “Primary Health Care” category (Table 1).”

Point 7.1: The description of article selection is confusing and appears inconsistent. It seems that a PUBMED search was used to identify related articles. For instance, at line 138, it stated “For selecting COVID-19-related articles, we used the list of all articles in the LitCovid 138 collection, i.e., 115314 articles as of April 7, 2021.” There may be journals identified from other databases that are not in PUBMED. Using PUBMED search only may miss articles from these databases.

Response 7.1:
As mentioned in Methods 2.3 LitCovid: A Database of COVID-19-related Literature from PubMed, LitCovid is a new open-resource literature hub developed by NCBI/NLM. It searches PubMed for COVID-19-related literature using the following search terms [1]:

“coronavirus”[All Fields] OR “ncov”[All Fields] OR “cov”[All Fields] OR “2019-nCoV”[All Fields] OR “COVID-19”[All Fields] OR “SARS-CoV-2”[All Fields]

Therefore, all articles in LitCovid are included in PubMed. At the beginning of this study, we manually counted the number of COVID-19 articles published in each journal. After learning the database of LitCovid, we conducted a comparison. We found that, for a small number of articles, we inevitably missed some of them when we looked at the article titles manually. Moreover, LitCovid provides a good classification of article contents, which allows us to further understand the content nature of the articles.

[1] Chen, Q., Allot, A., Lu, Z. Keep up with the latest coronavirus research. Nature 2020, 579, 193. doi:10.1038/d41586-020-00694-1

Point 7.2: On line 129, it states “The selection criteria were that the article must be published and indexed in PubMed in 2020 and that it must be in the English 129 language.” Then, what is the purpose of identifying relevant journals from other databases in the first place?

Response 7.2: For the journal search part, although PubMed provides a list of all indexed journals, it did not properly categorize the journals according to their professional content and properties. Although there was a classification of Primary Health Care in Broad Subject Terms for Indexed Journals, only journals included in MEDLINE were classified. The journals included in PMC were not categorized either. To allow the reader to more clearly determine the reasons for choosing different databases to collect family medicine journals, we have organized them in the Methods section. 2.1 Why Choose PubMed and Its Disadvantages as follows:

“Although PubMed provides a list of all indexed journals, it does not properly categorize them according to their professional content and nature. Although there is a classification of Primary Health Care in Broad Subject Terms for Indexed Journals, only journals indexed in MEDLINE are classified. The journals included in PMC are still not classified. Therefore, we collected family medicine journals in a more objective way using several existing classifications of journals from different databases. Moreover, because English is the most widely spoken language in the world, it has the greatest impact on the promotion of important knowledge. To understand the contribution of journals to the global epidemic, this study focused on English language journals and articles.”

Point 8: There seems to be inconsistency in the description. At line 141, it says, “we used the list of family medicine journals included in the previous section to find all articles in these journals…” There are 33 journals based on Figure 1. Then on page 147, it says, “in total, there were 25 SCI journals and 65 non-SCI journals.” How could that be if only the 33 journals were used to search for articles?

Response 8: Thank you for the careful evaluation. There is a missing word. It should be "there were 25 articles in SCI journals and 65 articles in non-SCI journals." The corrected statement is as follows:

“We searched all articles by title in LitCovid and recorded the authors, number of authors, time of PubMed inclusion, and LitCovid article category for each article. Moreover, we excluded non-English articles as well as articles published in PubMed in 2021; in total, there were 25 articles in SCI journals and 65 articles in non-SCI journals. Furthermore, we searched the articles in PubMed and recorded the country of the first author’s affiliation and searched the citation times of each article in Google Scholar.”

Point 9: It would be helpful to present the % of publications in each journal that are COVID-related. Only presenting the volume of publications during last year is not very informative.

Response 9: The number of articles published in each journal and the content, type, and presentation of the articles selected for publication are based on the business philosophy and motivation of the journal’s founder. Such an analysis of publishers was not the main purpose of this study. Therefore, these data and table were not presented in our study. However, the data and tables were organized for reviewers’ reference.

Please see attachment for the table.

The table is as follows:

Supplemental 1. Proportion of COVID-19 publication in family medicine journals indexed in PubMed in 2020

SCI abbr 1

Publication(n)

COVID-19 articles(n)

Proportion
(%)

Non-SCI abbr 1

Publication(n)

COVID-19 articles(n)

Proportion

(%)

Am Fam Physician

275

17

6.18

Afr J Prim Health Care Fam Med

110

29

26.36

Ann Fam Med

106

8

7.55

BJGP Open

125

24

19.20

Aten Primaria

12

2

16.67

Can J Rural Med

38

1

2.63

Aust J Gen Pract

195

49

25.13

Educ Prim Care

105

13

12.38

Aust J Prim Health

70

2

2.86

Fam Med Community Health

49

9

18.37

BMC Fam Pract

276

6

2.17

J Family Community Med

34

2

5.88

Br J Gen Pract

559

52

9.30

J Family Med Prim Care

1202

68

5.66

Can Fam Physician

208

18

8.65

J Gen Fam Med

114

6

5.26

Eur J Gen Pract

35

7

20.00

J Prim Care Community Health

215

39

18.14

Fam Med

146

3

2.05

J Prim Health Care

60

8

13.33

Fam Pract

163

10

6.13

J Rural Med

38

0

0.00

J Fam Pract

149

9

6.04

Korean J Fam Med

80

1

1.25

J Am Board Fam Med

162

3

1.85

Malays Fam Physician

43

3

6.98

NPJ Prim Care Respir Med

55

1

1.82

S Afr Fam Pract (2004)

84

8

9.52

Phys Sportsmed

99

2

2.02

Prim Care

56

2

3.57

Prim Care Diabetes

119

4

3.36

Prim Health Care Res Dev

64

2

3.13

Scand J Prim Health Care

61

1

1.64

1Abbreviation in PubMed.

Point 10: Line 226, it should be made clear that only publications in PUBMED were assessed.

Response 10: Thank you for the suggestion. We have completed the corrections according to your suggestions. The correction is as follows:
“In the 33 family medicine journals indexed in PubMed in 2020, 5107 articles were published, 409 of which were COVID-19-related articles. Original articles accounted for approximately one-fourth of the articles in the category. When classified based on article content, articles on prevention accounted for approximately 40% of the articles. At the beginning of the epidemic, family medicine journals published 10 articles in January 2020, accounting for >10% of all COVID-19-related publications worldwide. However, the proportion of COVID-19-related articles published in family medicine journals among all disciplines is extremely low for the entire year.”

Point 11: It would be very helpful to at least identify the major topics in the larger categories of publications such as prevention or diagnosis. Are they merely reporting how many were diagnosed, discussed innovative ways to diagnose patient, or identified issues in diagnosing patients in primary care setting? These would be more informative for improving primary care.

Response 11: In this study, the content classification of the articles mainly followed LitCovid’s existing classification and therefore did not further analyze the content. There are two main reasons for this: first, the content of articles often discusses different minor topics, and manual classification is often too subjective and difficult. Second, this is a bibliometric study, and the analysis of the content is more similar to the scope of review articles. Therefore, we did not further classify each article according to its content.

Reviewer 4 Report

The authors defined core conceptual elements to represent COVID-19 publications in family medicine journals. In summary, the idea is applicable but the novelty is too weak. Some comments and suggestions are given below.

(1) Please clearly describe the motivation of your paper in the Introduction Section of your paper.

(2) Please clearly describe the contributions of your paper in the Abstract, the Introduction Section and the Conclusions Section of your paper.

(3) Some typos and errors should be carefully checked. 

(4) The authors have not given detailed discussion in the experimental results part, they should give more comments on the results in the revised version. An additional discussion section should be included in this paper.

Author Response

Response to Reviewer 4 Comments

Attachment is the response to all reviewers.

The authors defined core conceptual elements to represent COVID-19 publications in family medicine journals. In summary, the idea is applicable but the novelty is too weak. Some comments and suggestions are given below.

Point 1: Please clearly describe the motivation of your paper in the Introduction Section of your paper.

Response 1: Thank you for the suggestion. In the face of the epidemic, the global research on COVID-19 is growing exponentially. The primary motivation for this study was to understand the research in the professional journals of family physicians, who are on the frontline of disease control, and to learn about the direction of improvement.

In the Introduction section, we mention that “This research and analysis approach provides a further comprehensive view of the contribution of family medicine to emerging infectious disease studies and its key role for combating such viruses.”

Understanding the situation before making improvements is the main motivation of this study.

Point 2: Please clearly describe the contributions of your paper in the Abstract, the Introduction Section and the Conclusions Section of your paper.

Response 2: We thank the reviewer for the suggestion. The contribution of this study is to understand the worldwide PubMed indexed research on COVID-19 in family medicine journals. It also examines the role of family medicine research in this epidemic. The results of this study revealed that although the initial research performance of family medicine journals was impressive, the intensity and continuity of research was not. This result is the main contribution of this study.

In the Abstract, we mention:
“At the beginning of the epidemic, 10 articles were published in family medicine journals in January 2020, accounting for 11% of all COVID-19-related articles worldwide; however, this accounted for <0.5% of all disciplinary studies in the entire year. Therefore, family medicine journals indeed play a sentinel role, and the intensities and timeliness of COVID-19 publications deserve further investigation.”

The Introduction section presented the following:
“Therefore, the present bibliometric study was conducted by collecting the publication patterns of global family medicine journals indexed in PubMed in 2020; analyzing the COVID-19-related articles in terms of the number of studies published, their categories, time of publication, and country where the research was conducted; and ranking them according to the most-cited and fastest-published articles. This research and analysis approach provides a further comprehensive view of the contribution of family medicine to emerging infectious disease studies and its key role for combating such viruses.”

The Conclusion section presented the following:
”The present study was conducted to understand the state of COVID-19-related publications in family medicine journals and to analyze the contribution and role of family medicine in the epidemic. In addition to the need to strengthen the intensity and continuity of COVID-19 articles published in these journals, as is the nature of family medicine, there should be more diverse research directions for this disease.”

In each of these paragraphs, we do present the main contributions of this study.

Point 3: Some typos and errors should be carefully checked. 

Response 3: Thank you for the reviewer’s correction. This article has been reviewed by a professional English editorial company, and the revision will be corrected again for typographical errors, grammar, and phrases.

Point 4: The authors have not given detailed discussion in the experimental results part, they should give more comments on the results in the revised version. An additional discussion section should be included in this paper.

Response 4: This study is a descriptive analysis study and no experiments were conducted. In our study, the classification into SCI and non-SCI was done mainly for the purpose of data collection. However, although the number of journals included in the non-SCI group was smaller, the number of COVID-19-related articles published in the non-SCI group was comparable to that of the SCI group, and the total number of COVID-19 citation articles even exceeded that of the SCI group. This also shows that although the impact factor or the number of citations can bring objective and quantitative data to the study, it cannot give a comprehensive evaluation of individual research content.

Based on the reviewer's suggestion, we have added the following discussion to this section:

“In this study, the classification into SCI and non-SCI was conducted mainly for data collection. However, although the number of journals included in the non-SCI group was smaller, the number of COVID-19-related articles published in the non-SCI group was comparable to that of the SCI group, and the total number of citations of COVID-19 articles even exceeded that of the SCI group. This also shows that, although the impact factor or number of citations can provide objective and quantitative data for research, it cannot give a comprehensive evaluation of individual research content.”

Round 2

Reviewer 3 Report

Thank you for responding to my comments. I don't have any additional comments.

Reviewer 4 Report

Revised manuscript is good. Thank you for working on your revised manuscript.